

# Optimizing inventory management: a causal inference-driven Bayesian network with transfer learning adaptation

Zhu Xi[1], Wei Guan[1,2] and Ahmet Savasan[3]

[1] Department of Auditing, School of Management, Wuhan Technology and Business University, Wuhan, China
[2] Hubei Research Center for Business Service Development, Wuhan Technology and Business University, Wuhan, China
[3] Department of Sports Science, Near East University, Nicosia, Cyprus

## ABSTRACT

Inventory management faces increasing challenges, including data limitations and demand uncertainty. To enhance inventory forecasting and optimization in supply chain management, this study proposes a Transfer-learning Bayesian Network (TBN) framework that integrates causal inference and transfer learning. Unlike traditional inventory forecasting models that rely on historical data patterns, the proposed framework introduces a causal inference-based Bayesian network to establish explicit causal relationships between sales volume, sales revenue, and inventory levels. To address data scarcity and improve generalization, a novel transfer learning mechanism is incorporated, leveraging a balanced weight coefficient method to optimize model adaptation from a source domain to a target domain. The results indicate that the proposd approach ensures effective knowledge transfer and maintains prediction accuracy with limited training data. The TBN model consistently outperforms traditional machine learning methods and other Bayesian-based models. On the self-constructed dataset, the TBN framework achieved a mean squared error (MSE) of 4.97 and a mean absolute error (MAE) of 2.78, demonstrating superior predictive accuracy. Additionally, an analysis of the balance weight coefficient further validated its role in enhancing transfer learning efficiency and model robustness, which provides a scalable and adaptable solution for intelligent inventory management in supply chain systems.

## INTRODUCTION

With the ongoing progression of globalization and the intensifying market competition, supply chain management (SCM) has become a vital component of contemporary enterprise strategy. Transitioning from traditional supply chain models to advanced intelligent supply chain management systems, SCM has undergone substantial development and transformation. The essence of the supply chain encompasses all links and activities involved in the journey of a product from production to delivery to the final consumer (*Almahdy et al., 2021*). This includes manufacturers, suppliers, logistics,

Corresponding author
Wei Guan, guanwei@wtbu.edu.cn

distributors, retailers, and ultimately, the end consumers. Effective SCM can markedly enhance operational efficiency, reduce costs, and improve customer satisfaction and market competitiveness. Within the modern supply chain management system, inventory management holds a pivotal role as the central link. The primary objective of inventory management is to ensure customer demand is met while minimizing inventory levels, thereby reducing costs and optimizing the use of funds. Effective inventory management enhances the overall efficiency of the supply chain and mitigates the risks and losses associated with inventory shortages or surpluses (*Asadkhani, Fallahi & Mokhtari, 2022*). The transition from traditional supply chain models to intelligent supply chain systems brings significant benefits, including improved operational efficiency through automation and real-time analytics, enhanced supply chain transparency *via* end-to-end data integration, reduced operational and inventory costs through process optimization, more accurate demand forecasting enabled by advanced predictive models, and faster response times supported by real-time monitoring and predictive alerts. These improvements allow enterprises to respond rapidly to market fluctuations, maintain optimal inventory levels, and ensure reliable product and service delivery, thereby enhancing customer satisfaction, strengthening brand reputation, increasing adaptability to market changes, and ultimately reinforcing overall market competitiveness and long-term sustainability.

However, inventory management encounters numerous challenges. Firstly, the uncertainty and volatility of market demand heighten the complexity of inventory management. Secondly, the increased complexity of supply chains due to globalization, coupled with the greater diversity and decentralization of supply chain links, necessitates the consideration of more variables and uncertainties in inventory management. Additionally, the heightened risk of supply chain disruptions, such as natural disasters, political unrest, and supplier bankruptcies, imposes greater demands on inventory management (*Chen, Liu & Zhang, 2022*). To address these challenges, modern inventory management has begun incorporating artificial intelligence and machine learning technologies to forecast future demand changes and optimize inventory decisions by analyzing historical sales data, market trends, and external environments. Machine learning algorithms, including regression analysis, time series forecasting, and classification models, can extract valuable insights from large datasets and identify potential patterns and trends, thereby providing more accurate and timely inventory forecasts (*Cheng et al., 2021*).

The improvement in the accuracy of inventory forecasting through artificial intelligence (AI) and machine learning primarily stems from data-driven pattern recognition, dynamic adaptability, and causal inference. Deep learning techniques can identify complex temporal patterns and capture nonlinear relationships, avoiding reliance on the linear assumptions of traditional statistical methods, thereby enhancing the precision of long-term forecasting. Reinforcement learning further optimizes inventory decisions based on real-time data, adapting to demand fluctuations and increasing supply chain resilience. Causal inference methods provide a unique advantage in uncovering cause–effect relationships between variables, allowing companies to distinguish genuine drivers of inventory changes from spurious correlations. For example, Bayesian networks can model

causal links between sales, promotions, and inventory turnover, enabling more reliable predictions and policy simulations in real-world scenarios. Transfer learning complements these capabilities by enabling models trained on one product category, season, or regional market to be efficiently adapted to others, significantly reducing the need for large amounts of new data and accelerating deployment in diverse operational contexts. This is particularly valuable for enterprises managing heterogeneous product portfolios or rapidly entering new markets. In practice, the Enterprise Resource Planning (ERP) system plays a crucial role in optimizing inventory management by integrating multisource data and intelligent analytics to improve inventory turnover, reduce excess stock, and prevent stockouts. It supports real-time inventory monitoring and leverages AI, including causal inference models, to predict demand fluctuations and inform precise replenishment strategies. Meanwhile, the Vendor Managed Inventory (VMI) model allows suppliers to directly manage inventory levels, leveraging transfer learning to adjust forecasting models to the supplier's product mix and demand patterns, thereby reducing supply chain disruption risks and further improving forecasting accuracy. Causal inference is essential in inventory optimization, as it analyzes the causal relationships between inventory, demand, and supply chain factors, identifying key influences to optimize replenishment strategies and enhance supply chain resilience. By combining ERP systems, VMI, and causal inference methods, businesses can achieve more precise inventory control, improve supply chain efficiency and market competitiveness, and realize refined operations. ERP systems enhance inventory management by integrating real-time data across procurement, production, and sales, enabling more accurate demand forecasting and coordinated replenishment. This improves inventory turnover by aligning stock levels with actual market needs, reduces excess stock through better visibility and control, and mitigates common issues such as stockouts or overstocking, thereby increasing overall supply chain efficiency.

Many organizations grapple with the issue of limited data volume in real-world applications, leading to the challenge of insufficient data for training deep learning models. To mitigate this problem, transfer learning techniques offer a viable solution. Transfer learning enhances prediction performance with limited data by transferring the knowledge from pre-trained models in the source domain to the target domain (*Dag et al., 2023*). Consequently, this article integrates existing causal inference Bayesian networks with transfer learning methods to optimize inventory management in supply chain management, aiming to enhance the efficiency of intelligent optimization. The specific contributions are as follows:

1. Addressing the need for inventory optimization in supply chain management, the existing Bayesian network model is enhanced, and a TBN inventory management optimization framework based on causal inference and transfer learning is established by introducing a transfer learning mechanism, aiming to achieve high-precision inventory forecasting and optimization.

2. The constructed TBN framework achieves highly accurate inventory prediction using a public dataset, with results significantly outperforming unimproved traditional methods.

3. In practical inventory management applications, the challenge of limited data is addressed by balancing the parameters of the source and target domains through a method of balancing weight coefficients. It is demonstrated that the TBN framework can achieve high-precision inventory prediction and optimization even with a self-built dataset, as evidenced by comparisons with improved Bayesian network methods such as Tree-Augmented Naïve Bayes (TAN) and Weighted Averaging Tree-Augmented Naïve Bayes (WATAN).

The rest of the article is organized as follows: 'Related Works' reviews related works on inventory management optimization and causality research. 'Methodology' presents the proposed TBN framework. 'Experiment Result and Analysis' provides the experimental details and results. Finally, the conclusion is drawn.

# RELATED WORKS

## Inventory management and optimization studies

To ensure the continuity of sales, it is necessary to stock some goods in reserve to cope with potential changes. This buffer stock helps balance demand fluctuations and production adjustments, which necessitates maintaining a certain inventory level to ensure the presence of safety stock, originating from the theory of safety stock (*Forney & Mueller, 2022*). A higher safety stock reduces the likelihood of stock-outs but increases surplus stock; thus, an optimal safety stock level must be maintained within a reasonable range. This level can be determined from consumer demand or expected service levels, and quantitatively established by examining demand changes (*Goldberg, Reiman & Wang, 2021*). Harris proposed the Economic Order Quantity (EOQ) model to control inventory costs; the EOQ model is an inventory management method aimed at optimizing operational efficiency (*Ivanov, Tsipoulanidis & Schönberger, 2021*). It addresses mismatches between production and demand within enterprises, focusing on the rational allocation of inventory costs. *Chung et al. (2021)* further refined the EOQ model for perishable products (*Kiciman & Sharma, 2019*). With the evolution of supply chain theory, traditional inventory control methods have continuously improved. Alongside analyzing and researching the EOQ model, the classic inventory control method based on the Pareto principle—ABC analysis—has been utilized. ABC analysis segments products hierarchically, recognizing that many products follow this distribution (*Kitson et al., 2023*). With advancements in computer information technology, *Goldberg, Reiman & Wang (2021)* have enhanced the theory of multilevel inventory, promoting the development and application of supply chain management concepts in inventory management. They proposed two multilevel inventory control models (*Luo et al., 2016*), which have since been widely adopted across various industries. The ERP system, which starts with supply chain management, effectively improves inventory turnover and alleviates inventory problems. Subsequently, the Vendor Managed Inventory (VMI) model emerged, employing

just-in-time delivery technology to prevent stock-outs and enable downstream suppliers to accurately predict enterprises' purchasing needs (*Ma & Fildes, 2021*). The VMI model, based on joint inventory management, has proven effective in supply chain inventory management strategies, gaining academic recognition and widespread implementation. VMI improves supply chain performance by allowing suppliers to directly monitor and replenish inventory based on real-time sales and demand data. This proactive approach reduces the risk of stock-outs, ensures timely product availability, and enhances the accuracy of inventory forecasting through closer alignment between supply and actual consumption patterns. Traditional inventory control methods typically use fixed order quantity (EOQ), reorder point (ROP), or safety stock strategies, relying primarily on historical data and statistical models to determine inventory replenishment rules. These methods are relatively static and struggle to adapt to demand fluctuations in complex supply chains. In contrast, the multi-echelon inventory model focuses on optimizing inventory across different supply chain levels (such as suppliers, warehouses, and retailers). By leveraging information sharing, demand forecasting, and dynamic inventory allocation, it enhances overall supply chain efficiency, reduces total inventory costs, and minimizes stockout risks, making inventory management more flexible and adaptive.

## Causal reasoning studies

A causal inference model is an algorithm designed to explore causal relationships between variables. Unlike the latent causation framework (*Mishra, Wu & Sarkar, 2021*) commonly employed in economics and sociology, causal inference models in computer science often utilize the structural causation framework. Typically, these models employ a Bayesian network (*Ntakolia et al., 2021*) as the foundation to construct the causal structure between variables, thus also known as causal Bayesian networks or causal networks. *Pournader et al. (2021)* introduced an innovative strategy to transform the constrained problem of a directed acyclic graph (DAG) into a sequential optimization problem on a matrix of real numbers, addressing the limitations of combinatorial optimization. This smooth representation was incorporated into the scoring function of linear structure learning, resulting in the Non-combinatorial Optimization via Trace Exponential and Augmented (NOTEARS) algorithm, which accurately learns the graph structure from continuous variables. Building on this, numerous research efforts have optimized the algorithm with structural constraints, applying it to nonlinear systems or ensuring algorithmic consistency. *Pramodhini et al. (2023)* developed a maximum likelihood function as a scoring function based on the Linear Non-Gaussian Acyclic Model (LiNGAM) model, proposing the Bayes-LiNGAM algorithm to estimate the causal structure. To achieve strong artificial intelligence, many scholars have leveraged causal theory to design algorithms that help machines learn causal relationships from data, aiming for stable predictions with high accuracy and interpretability. *Sutejo, Suprayitno & Latunreng (2023)*, from a data-driven perspective, proposed a framework to automatically collect causal relationships from a large web *corpus*, which quantifies the strength of causal relationships between various texts, addressing the common-sense causal reasoning problem. *Tadayonrad & Ndiaye (2023)* introduced the StableNet method, which

eliminates dependencies between features by learning the weights of training samples, thereby focusing more on discerning the true connections between features and labels. This method utilizes genuinely relevant causal features for prediction, achieving stable performance in non-smooth environments. *Wu, Wang & Wu (2022)* applied Bayesian network to investigate qualitative and quantitative causal relationships between context and quality of service (QoS) metrics, deploying their findings in a real multimedia conferencing system. Causal reasoning enables inventory management systems to move beyond simple correlations by uncovering the true cause–effect relationships between inventory levels, customer demand, and broader supply chain dynamics. By identifying how changes in demand patterns, lead times, or supplier performance directly influence stock levels, causal models allow managers to simulate potential interventions and predict their outcomes with greater reliability. This deeper understanding supports more optimized inventory decisions, such as adjusting safety stock, prioritizing replenishment, or reconfiguring supply chain flows to balance service levels and costs.

The aforementioned study underscores the importance of maintaining safety stock for enterprises to manage demand fluctuations and production adjustments effectively. The essence of safety stock theory is to balance these fluctuations and adjustments, thereby mitigating the risks of stock-outs and excess inventory. With advancements in supply chain theory, multilevel inventory control methods and ERP systems have matured, significantly enhancing the efficiency and precision of inventory management. In the current era of data science, a substantial volume of data is essential for achieving high-accuracy inventory analysis. Causal reasoning, distinct from the traditional latent causal framework of economics and sociology, emphasizes the structural causal framework. This approach constructs the causal structure between variables, enabling precise causal relationship analysis. Consequently, optimizing inventory management through causal reasoning can uncover the intricate causal relationships between inventory, demand, and various supply chain links. This understanding facilitates the development of more scientific and efficient inventory management strategies.

# METHODOLOGY

## Bayesian networks

Causal inference, particularly when implemented through Bayesian networks, enables the explicit modeling of directional dependencies between sales and inventory, distinguishing true causal effects from mere correlations. By capturing how changes in sales volume directly influence inventory levels—and *vice versa*—Bayesian networks support more accurate forecasting and allow scenario-based simulations for optimized inventory decisions. There is a robust relationship between Bayesian networks and causal inference models. Bayesian networks utilize DAGs to represent conditional dependencies between variables, serving as a specific form of causal inference modeling. When a Bayesian network not only represents probabilistic dependencies but also explicitly expresses causal relationships, it is termed a causal Bayesian network. These networks are capable of performing probabilistic reasoning as well as causal reasoning, such as calculating intervention effects and counterfactual reasoning. Causal inference models construct

causal structures between variables using Bayesian networks and leverage data to learn these structures and parameters, achieving highly accurate causal analysis and prediction (*Yamayoshi, Tsuchida & Yadohisa, 2020*).

A Bayesian network consists of a directed acyclic graph (DAG) and a set of conditional probability distributions (CPDs). Suppose the Bayesian network contains $n$ random variables $X_1, X_2, \ldots, X_n$, and their joint probability distribution is denoted as.

$$P(X_1, X_2, \ldots, X_n) = \prod_{i=1}^{n} P(X_i | \mathrm{Pa}(X_i)) \tag{1}$$

where $\mathrm{Pa}(X_i)$ denotes the set of parent nodes of node $X_i$. The conditional probability of each node $X_i$ is represented as the conditional probability distribution of the given parent node set $\mathrm{Pa}(X_i)$ $P(X_i | \mathrm{Pa}(X_i))$. These conditional probability distributions are derived through data learning and are typically represented in the form of probability tables. The edge probabilities for any subset of variables are computed from the joint probability distribution. For example, the marginal probability of variable A can be expressed by Eq. (2):

$$P(X_A) = \sum_{X_{\setminus A}} P(X_1, X_2, \ldots, X_n) \tag{2}$$

where $X_{\setminus A}$ denotes all variables except $X_A$.

Bayes' theorem is commonly used in Bayesian networks for probability updating and inference. Given the observed evidence E, compute the posterior probability of some variable of interest X:

$$P(X|E) = \frac{P(E|X)P(X)}{P(E)} \tag{3}$$

where, $P(E)$ can be obtained by marginalization

$$P(E) = \sum_{X} P(E|X)P(X). \tag{4}$$

This enables the updating and inferring of the probability distribution of the variable of interest given new evidence. Bayesian networks efficiently represent and reason about conditional dependencies and causal relationships in complex systems, and are widely utilized across various research domains involving inference.

In practical applications, the reasoning process of causality is critically important. For this article, considering market sales factors and inventory situations in the context of management optimization, the general causal path topology can be illustrated as shown in Fig. 1.

That is, causality in this article is primarily constructed based on sales volume. This approach differs from traditional research, which typically relies on historical inventory changes for data analysis. By focusing on changes in causality, the model's performance can be analyzed more effectively.

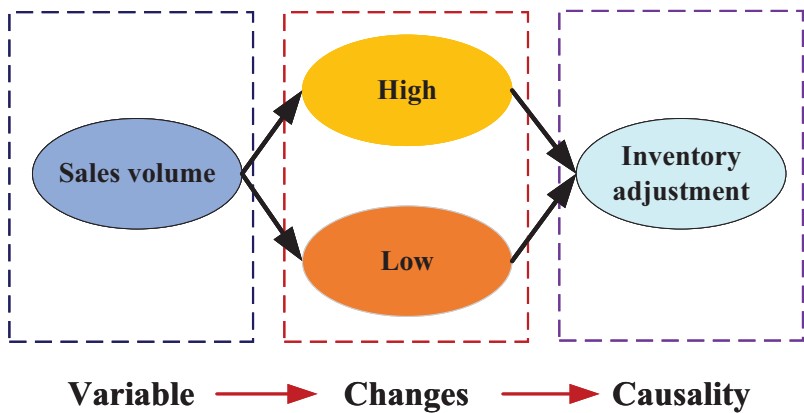

**Figure 1 The causal relationship of inventory changes.**

Most current research on Bayesian network classifiers emphasizes learning the network topology from training data. To study the dependencies between the learned attributes in the network, this article employs the log-likelihood function to conduct an inference study on the causal relationships. On the basis of the known probability distribution function introduced in Eq. (1), we have a further log-likelihood function calculation, given the dataset $D = \{D_1, D_2, \ldots, D_m\}$, the log-likelihood function is:

$$\mathcal{L}(\theta; D) = \sum_{j=1}^{m} \log P(D_j|\theta) \tag{5}$$

where $\theta$ denotes the model parameters and $P(D_j|\theta)$ denotes the probability of a data point $D_j$ under a given parameter $\theta$. The log-likelihood function is used to calculate the log-likelihood value of the model thus enabling the assessment of the merits of different causal models. The Bayesian network approach offers advantages over traditional time series methods (*e.g.*, Autoregressive Integrated Moving Average (ARIMA), long short-term memory (LSTM)) in forecasting accuracy and training time for inventory prediction. Firstly, Bayesian networks can capture causal relationships between inventory, sales, and supply chain factors, rather than relying solely on historical data-based linear or nonlinear patterns, improving interpretability and prediction accuracy. Secondly, Bayesian networks use probabilistic inference, allowing effective predictions even with limited or incomplete data, whereas time series methods typically require large datasets for training. Additionally, Bayesian networks have lower computational complexity and shorter training time, making them well-suited for real-time inventory management and dynamic decision optimization.

### Transfer learning methods

Transfer learning is a machine learning method that aims to utilize knowledge gained from one task to improve performance on a related task. Unlike traditional machine learning methods, transfer learning leverages existing knowledge to enhance the learning efficiency and effectiveness of new tasks, particularly when data is scarce or costly to acquire, offering

a significant advantage. The fundamental concept of transfer learning involves learning models from the source domain and source task, then transferring this knowledge to the target domain and target task (*Zekhnini et al., 2021*). Transfer learning mitigates the limited data challenge in deep learning–based inventory optimization by leveraging pretrained models from related domains to initialize network parameters with learned feature representations. This approach reduces the reliance on large volumes of task-specific data, accelerates convergence, and improves model generalization, enabling accurate inventory predictions even in data-scarce scenarios. The crux of transfer learning lies in effectively sharing and adapting knowledge from the source task to address challenges in the target task. This often involves migrating some or all of the trained model parameters from the source task to the target task, a method commonly employed in neural network research. In transfer learning, a combination of source task loss $\mathcal{L}_S$ and target task loss $\mathcal{L}_T$ is usually introduced. The goal is to minimize the combined loss function:

$$\mathcal{L} = \mathcal{L}_T + \lambda \mathcal{L}_S \tag{6}$$

where $\lambda$ is the hyperparameter that weighs the source task loss and the target task loss. Suppose the source task model parameters are $\theta_S$ and the target task model parameters are $\theta_T$. Transfer learning is accomplished by fine-tuning the source task parameters by Eq. (7).

$$\theta_T = \theta_S + \Delta\theta \tag{7}$$

where $\Delta\theta$ denotes the tuning of the model parameters in the target task. Transfer learning, in scenarios where data is scarce or costly to obtain, leverages knowledge from similar tasks or domains, reducing reliance on large amounts of labeled data. This enhances the model's generalization ability, making inventory forecasting more efficient and adaptable to different business environments.

## The establishment for the transfer-learning Bayesian network

Considering the inventory management requirements within the framework of causal reasoning addressed in this article, we utilize Bayesian networks and transfer learning techniques for network design. This approach is based on the dual integration of sales volume and historical inventory data to achieve more precise optimization of inventory items. During the Bayesian network parameter transfer process, it is assumed that the target network in the target domain consists of parameters to be estimated. It is further assumed that there are N resource domains comparable to the target network. Any network comparable to the target network is referred to as a resource network, and these networks can provide valuable information for learning the target network. The TBN network constructed is illustrated in Fig. 2.

The target network can be expressed as $\Delta^T = \left\{D^T, G^T, V^T\right\}$, where the data of the target network is characterized by $D^T$, the structure of the target network is characterized by $G^T$, and the dimension of the target network is expressed by $V^T$. The resource network is expressed as $\Delta^S = \left\{D^S, G^S, V^S\right\}$, where the data of the resource network is represented

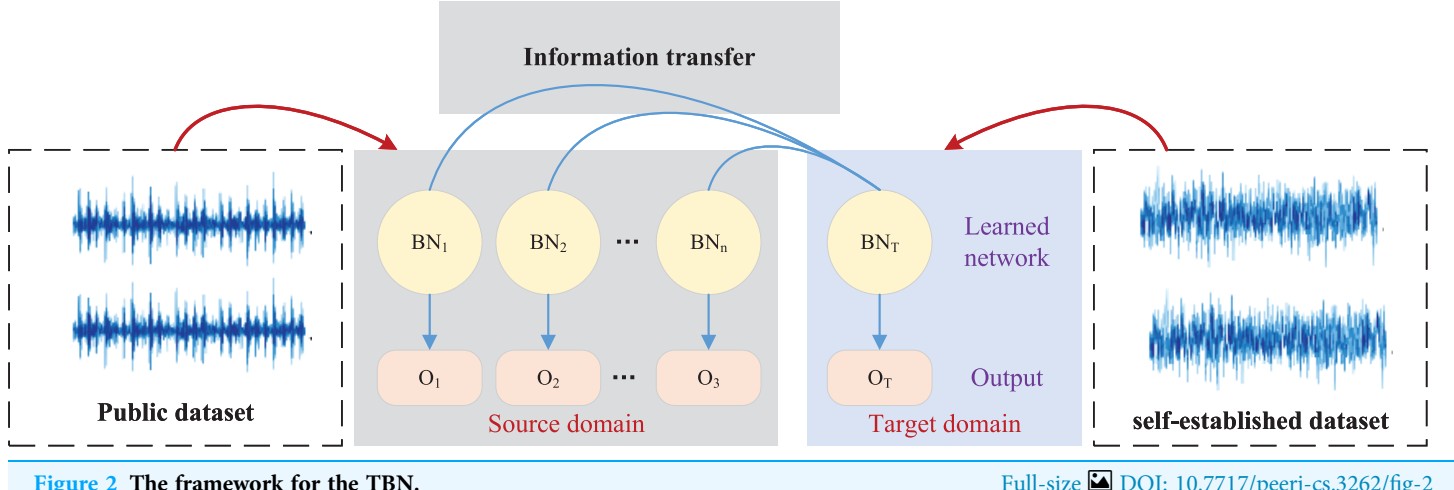

**Figure 2 The framework for the TBN.**

by $D^S$, the structure of the resource network is characterized by $G^S$, similar to the dimension of the target network, and the dimension of the resource network is expressed by $V^S$. The corresponding data under the public dataset are analyzed, followed by parameter transfer according to the characteristics of Bayesian networks.

Under the framework illustrated in Fig. 2, this article first normalizes the sales degree and sales quantity data in the sales process, which are then fed into the Bayesian network. The model's output is the remaining inventory for the day, enabling an assessment of the inventory status. To ensure the accuracy of the model parameters during the transfer process, we employ a parameter adjustment method as exemplified in Eq. (7) for parameter weighting adjustment. The migrated parameters can be expressed by Eq. (8):

$$\theta_T = \alpha\theta_S + (1-\alpha)\Delta\theta \tag{8}$$

where $\alpha$ represents the balance weight coefficient, the magnitude of its value varies in [0, 1], which indicates the balance between the parameters concerning these two domains. The optimal transfer of the model is realized after adjusting the size of $\alpha$. Thus, it realizes the sales degree and sales quantity assessment based on the actual sample in the real scenario.

# EXPERIMENT RESULT AND ANALYSIS

## Dataset and experiment setup

After constructing the model, we proceed with testing and performance analysis. Initially, we evaluate the model using a publicly available dataset. For this study, we have selected the Retail Sales Forecasting dataset (*Zhang, Jiang & Li, 2022*), which contains extensive historical sales data. This dataset originates from major retailers in Brazil, with data transformations applied to ensure anonymity. In total, it comprises over 900 data points from multiple retailers, offering a robust foundation for analyzing overall inventory fluctuations and trends. A typical data sample from the dataset is illustrated in Fig. 3.

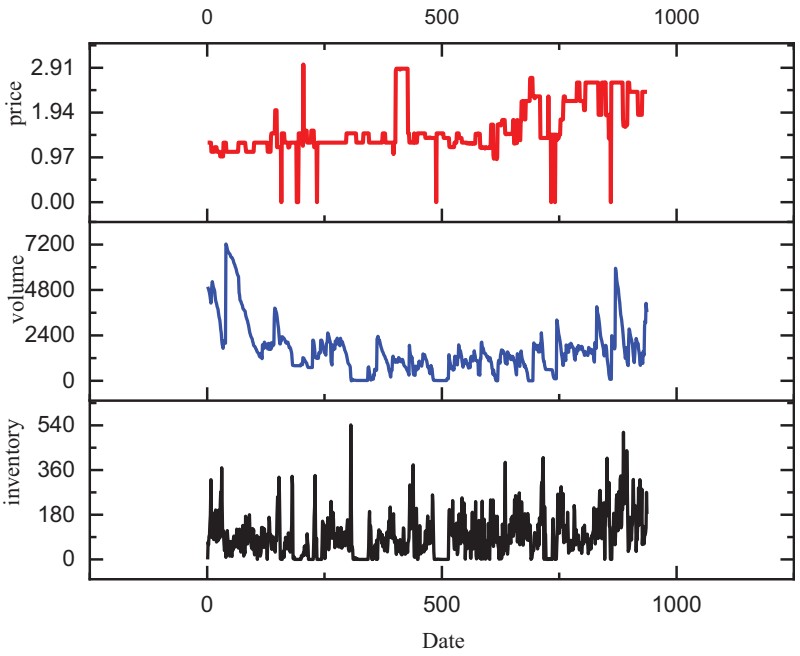

**Figure 3 The changes in product information according to the employed dataset.**

**Table 1 The experiment environment.**

| Environment | Information |
| --- | --- |
| CPU | I5-13500 |
| GPUs | GTX02080 |
| Language | Python 3.4 |
| Framework | Scikit-learn |

After confirming the use of the public dataset, we proceeded with the construction of the experimental environment. The specific details of the experimental environment are shown in Table 1.

In this article, considering that the final inventory forecasting analysis is a fitting analysis problem, we evaluate the model's performance primarily using the mean squared error (MSE) and mean absolute error (MAE) metrics. MSE is more sensitive to large errors, making it particularly effective for inventory forecasting.

$$\text{MSE} = \frac{1}{n} \sum_{i=1}^{n} \left( y_i - \hat{y}_i \right)^2. \tag{9}$$

MAE is insensitive to outliers and is more robust, providing a balanced measure of prediction accuracy.

$$\text{MAE} = \frac{1}{n} \sum_{i=1}^{n} \left| y_i - \hat{y}_i \right| \tag{10}$$

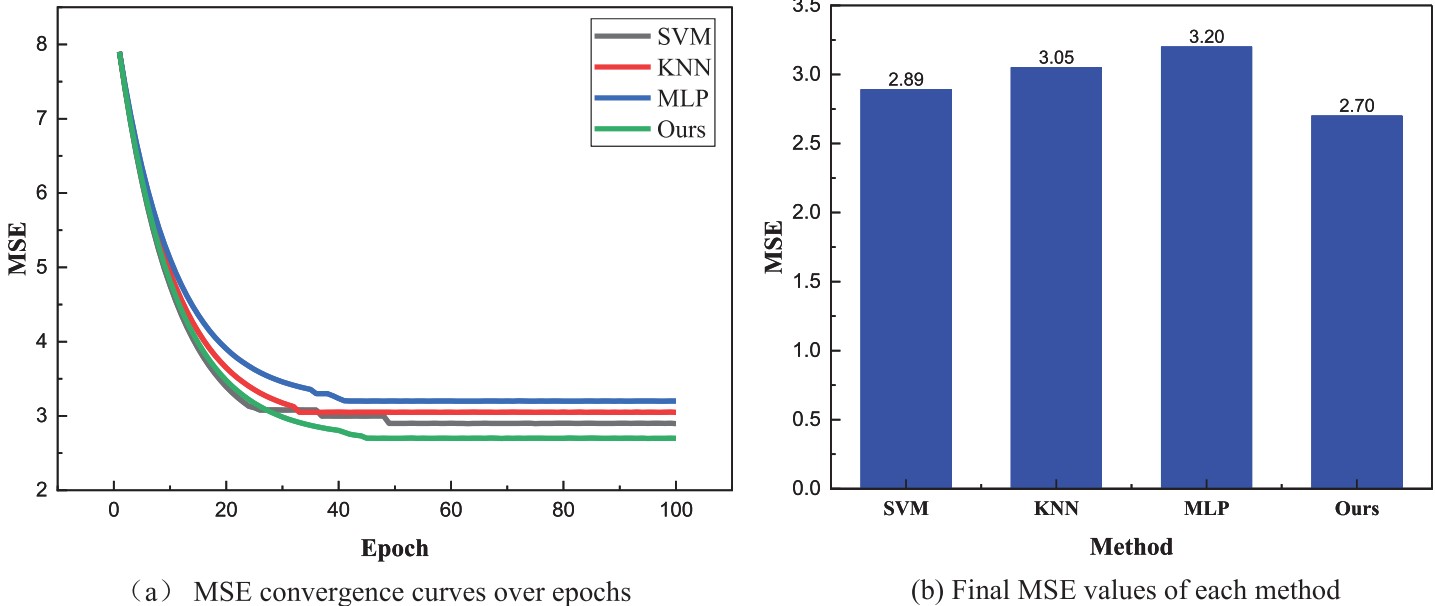

(a) MSE convergence curves over epochs      (b) Final MSE values of each method

**Figure 4** **The training process and MSE result.** (A) MSE convergence curves over epochs; (B) Final MSE values of each method.

where $y_i, \hat{y}_i$ represents the predicted and actual values, respectively. The two evaluation metrics, MSE and MAE, are used for more accurate model evaluation. In our method comparison, we analyze traditional time series machine learning methods such as MLP, SVM, and KNN, alongside Bayesian causal inference methods, including TAN (*Zhang et al., 2021*), WATAN (*Zheng, 2020*), and TAODE (*Zhuang et al., 2020*). Since this article focuses on enhancing Bayesian inference performance using the transfer learning approach, only the basic methods are compared on the public dataset. Meanwhile, the migrated TBN is compared with traditional TAN and other improved Bayesian methods on the self-constructed dataset.

## Model comparison and result analysis

After finalizing the training dataset and comparison methods, we proceeded with the model training and subsequent result analysis based on the public dataset using different methods. The results are illustrated in Fig. 4.

In Fig. 4, it is evident that due to the simpler data, only shorter and lower-dimensional time series methods have achieved more stable results after approximately 40 iterations. The Bayesian network method used has the smallest MSE of only 2.7. In contrast, the MLP method does not show a better advantage over longer iterations, likely due to falling into local extrema, with its final MSE stabilizing around 3.2. To further analyze the different models, we also compared their final MAE and training stabilization times, as shown in Fig. 5.

In Fig. 5, we present a comparison between the MAE and training time for the different models. The figure indicates that the overall trend for MAE is similar to

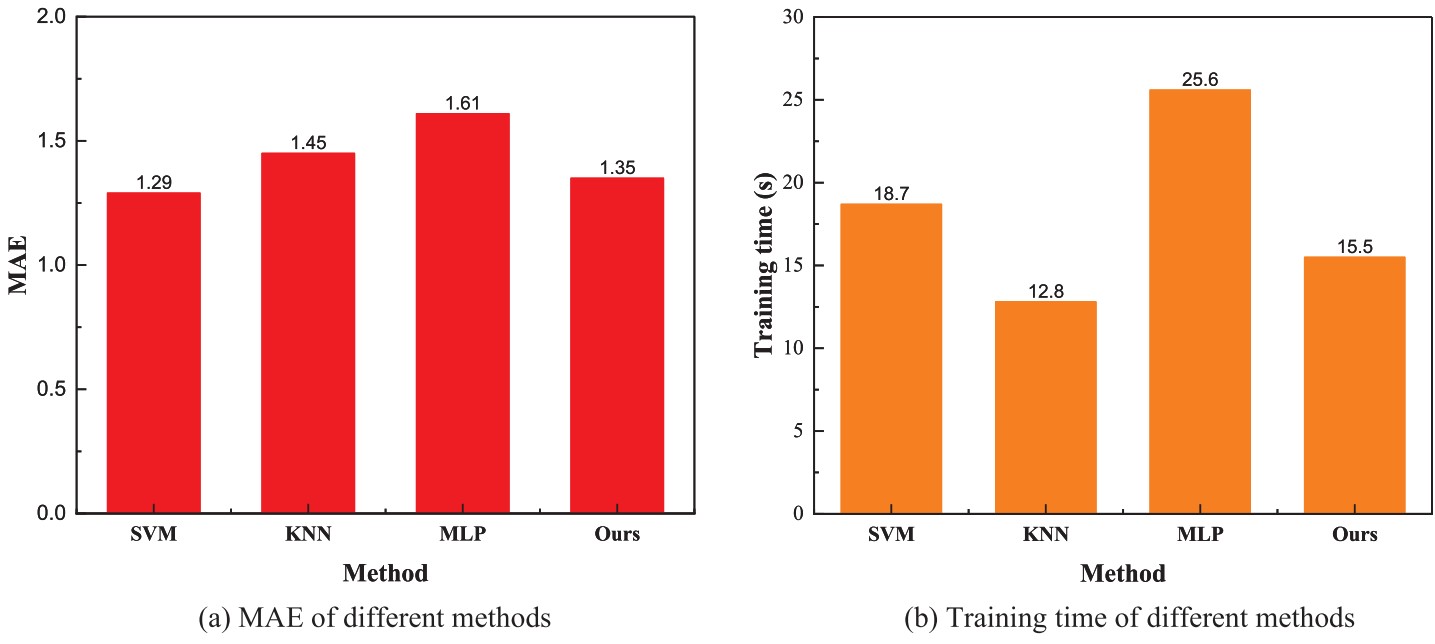

(a) MAE of different methods        (b) Training time of different methods

**Figure 5** **(A) MAE and training time (B) for the model comparison result.**

that of MSE, with both metrics showing a relatively average performance across the models. In terms of running time, network-based methods exhibit significantly higher training times compared to the simplest KNN method. Notably, the training efficiency of the Bayesian network method proposed is superior to that of the MLP method, offering faster speeds and higher accuracy. Consequently, model transfer through this method results in more efficient and precise inventory prediction and analysis.

## Model transfer analysis concerning self-established dataset

After completing the test on the public dataset, we find that the Bayesian network approach based on causal inference achieves higher accuracy in inventory prediction by combining historical inventory data with key information on sales volume and sales value. To further validate the rationality of the TBN framework proposed, we migrated the Bayesian network using the transfer approach detailed in 'The Establishment for the Transfer-Learning Bayesian Network' and conducted an analysis on the self-constructed database. In transfer learning, resource networks play a pivotal role in parameter estimation by leveraging pretrained model weights and shared feature representations from related tasks or domains. This enables the target model to initialize parameters closer to their optimal values, reducing the need for large volumes of task-specific data and accelerating convergence. However, network-based models, particularly deep architectures, often require longer training times compared to simpler methods such as k-neural network (KNN), due to their higher parameter complexity and iterative optimization processes. While this increased computational cost can be a limitation, it is generally offset by the superior generalization performance and adaptability achieved

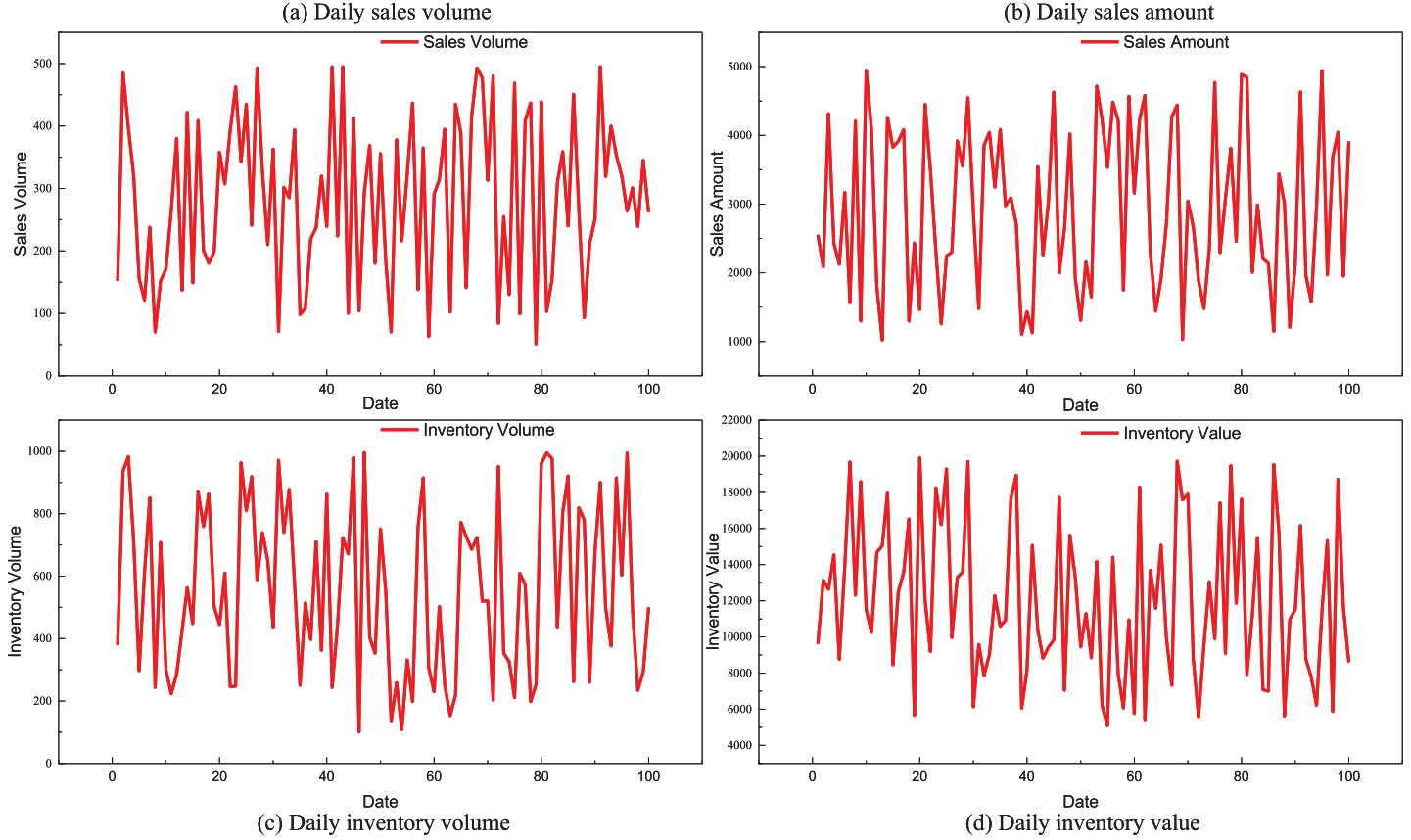

**Figure 6 Changes in product information in self-established dataset.** (A) Daily sales volume, (B) Daily sales amount, (C) Daily inventory volume, (D)Daily inventory value.

through transfer learning in complex inventory forecasting and supply chain contexts. Transfer learning enhances Bayesian causal models by enabling the reuse of structural priors and parameter estimates learned from related datasets, thereby providing a strong initialization for model training on self-constructed datasets. This approach mitigates the limitations of small sample sizes by transferring domain-relevant causal dependencies and probability distributions, which accelerates convergence and improves the stability and accuracy of causal relationship estimation. As a result, the model can more reliably uncover and quantify the links among sales, demand fluctuations, and inventory levels, even when the target dataset is limited in scale. The self-built database primarily records sales, sales volume, inventory, and inventory value data for specific product categories in large regional shopping malls. The model is trained and tested according to the requirements of the corresponding transfer framework. The four categories of features for a particular product in our constructed database are illustrated in Fig. 6.

Based on the previously constructed transfer framework, it is evident that the task undertaken involves fitting the current inventory analysis based on sales volume, sales degree, and historical total inventory value. This approach aims to implement the sequence-to-sequence regression method. The comparison methods used have been

**Table 2 The MSE and MAE comparison on the self-established dataset.**

| Methods | MSE | MAE |
| --- | --- | --- |
| Ours | 4.97 | 2.78 |
| TAN | 6.13 | 4.93 |
| WATAN | 6.83 | 5.16 |
| TAODE | 7.11 | 6.39 |

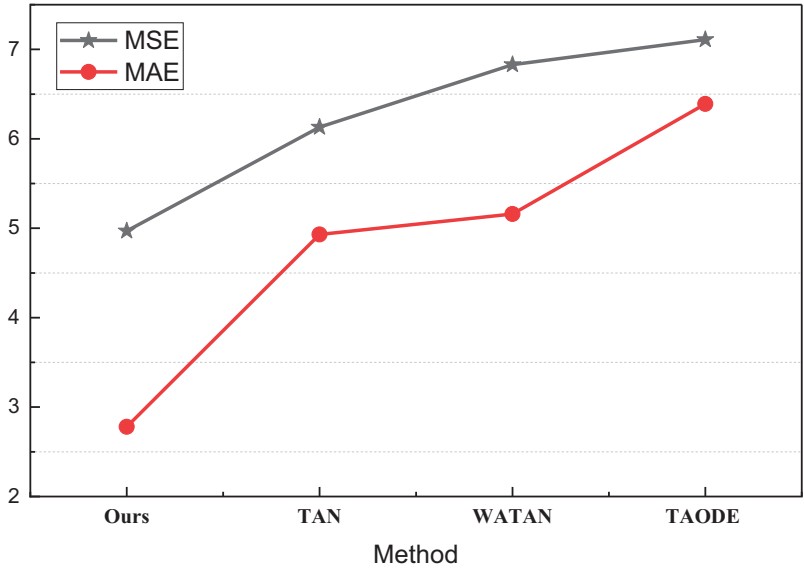

**Figure 7 The MSE and MAE comparison on the self-established dataset.**

described in 'Dataset and Experiment Setup'. To compare the effectiveness of the Bayesian network under transfer, we use the same MSE and MAE metrics. The results are shown in Table 2 and Fig. 7.

Through the comparison of different Bayesian networks, it can be seen that under the simpler model, we can form a better prediction effect on the smaller dataset used through certain transfer of parameters. TBN, the method proposed, has a better prediction effect. Compared to the single Bayesian network, the overall MSE of the transfer learning method can be better in the limited dataset. The MSE and MAE of the proposed method are 4.97 and 2.78, respectively, which are significantly better than those of TAN and other methods that make changes in the Bayesian network itself, *i.e.*, better inventory prediction is realized under the simpler causal inference model. To better analyze the effect of the transfer model for causal inference Bayesian network-like methods, we analyzed the transfer balance coefficient $\alpha$ in Eq. (8), and the results of MSE and MAE under different values are shown in Table 3 and Fig. 8.

In the TBN framework, balanced weight coefficients play a critical role in enhancing transfer learning across domains by controlling the contribution of source and target domain knowledge during model training. Properly tuned coefficients prevent

**Table 3  MSE and MAE among different α.**

| α | 0.1 | 0.2 | 0.3 | 0.4 | 0.5 | 0.6 | 0.7 | 0.8 | 0.9 |
|---|---|---|---|---|---|---|---|---|---|
| MSE | 6.59 | 6.71 | 5.11 | 5.13 | 5.01 | 4.97 | 5.21 | 5.35 | 6.09 |
| MAE | 3.11 | 3.29 | 3.07 | 3.15 | 3.05 | 2.78 | 3.13 | 3.47 | 3.59 |

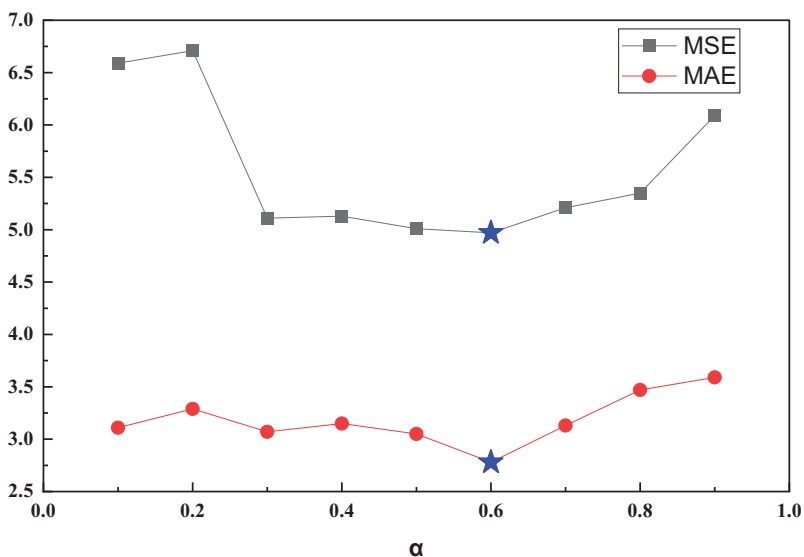

**Figure 8  MSE and MAE among different α.**

over-reliance on the source domain while ensuring sufficient knowledge transfer, thereby improving model adaptability, reducing negative transfer effects, and enhancing predictive accuracy in cross-domain inventory forecasting. In Fig. 8, we observe that with the increase of α, both MSE and MAE exhibit a decreasing trend, reaching the optimal transfer effect near α. At this equilibrium coefficient, the model achieves an MSE of 4.97 and an MAE of 2.78. This indicates that in future applications, the optimal transfer equilibrium coefficients can be determined by searching for the optimal balance under simpler data conditions, thereby enhancing the model's performance in practical applications.

## DISCUSSION

This article focuses on the optimization of inventory management in supply chain research and proposes a TBN inventory management optimization framework based on causal inference CBN and transfer learning. The framework utilizes causal inference to analyze the relationships between key variables such as inventory, sales volume, and sales revenue, improving inventory management accuracy and decision interpretability. Specifically, the framework first normalizes sales volume and sales revenue data before inputting them into the Bayesian network for modeling. The model outputs the remaining inventory for the day, enabling inventory status evaluation and replenishment optimization. During the parameter transfer process, to ensure model accuracy and adaptability across different

business scenarios, a balanced weight coefficient is introduced to dynamically adjust parameters between the source and target domains, optimizing the transfer effect. This method effectively reduces inventory overstock and stockout risks, enhances inventory forecasting accuracy, and improves the intelligence and competitiveness of supply chain management. Causal inference Bayesian networks provide greater explanatory power and predictive accuracy compared to traditional machine learning methods such as support vector machine (SVM), multilayer perceptron (MLP), and KNN. Inventory management involves complex causal relationships between factors like demand and supply, which traditional methods capture using large data volumes and complex model structures. However, SVM, MLP, and KNN methods function as "black-box" models, making it difficult to reveal causal relationships between variables and often perform poorly with insufficient or unevenly distributed data. Bayesian networks offer significant advantages in causal inference models by representing variables and their conditional dependencies in a structured, probabilistic framework. This allows for the explicit quantification of the strength and direction of causal relationships, enabling both predictive and explanatory analysis. Their graphical nature facilitates the integration of expert knowledge with empirical data, supports reasoning under uncertainty, and allows scenario simulation to assess the potential impact of interventions, thereby providing deeper contextual understanding of complex systems such as sales–inventory dynamics. In contrast, traditional machine learning methods lack causal explanation capabilities, limiting their effectiveness in complex supply chain environments. The TBN framework, through transfer learning, achieves balanced adjustment between resource and target domain parameters by introducing balanced weight coefficients, ensuring model accuracy and robustness across different domains. Compared to traditional methods like TAN and WATAN, the TBN framework offers superior performance in the transfer process. Although TAN and WATAN address causality modeling to some extent, they are vulnerable to differences between the source and target domains during transfer learning, leading to degraded model performance. Integrating historical inventory data with sales volume in a causal inference framework improves prediction accuracy by jointly capturing stock dynamics and demand drivers. This approach disentangles the causal impact of sales on inventory levels, enabling more reliable demand forecasts and optimized replenishment strategies.

The TBN framework introduces balanced weighting coefficients to maintain model simplicity while effectively adjusting and optimizing parameters between source and target domains. This approach enables higher prediction accuracy and adaptability within a simpler network structure. By combining the explanatory power of causal inference with the flexibility of transfer learning, the TBN framework provides a superior solution for complex inventory management scenarios. Inventory optimization is a crucial component of supply chain management. While traditional methods often require extensive historical data and complex model tuning, the TBN framework can maintain high prediction accuracy with less data or significant changes in data distribution through a combination of causal inference and transfer learning. This provides organizations with a flexible and

efficient inventory management solution in a dynamically changing market environment. Firstly, through causal inference Bayesian networks, we can more accurately assess the remaining state of inventory and optimize inventory management strategies, thereby reducing the risk of stockouts and excess inventory. This not only improves the overall efficiency of the supply chain but also reduces inventory management costs and enhances the competitiveness of the enterprise. Secondly, the introduction of transfer learning makes the TBN framework more adaptable across different domains. Whether it involves developing new markets or managing different product lines, the framework can be quickly deployed and optimized through effective parameter adjustment. Beyond inventory management, the application of the TBN framework in supply chain management is equally promising. With the continuous development of big data and artificial intelligence technologies, enterprises will be able to collect and process more diverse and massive amounts of data, further enhancing the TBN framework's applicability. By integrating with Internet of Things (IoT) technology, real-time monitoring of inventory and sales data, and applying causal reasoning and transfer learning to more complex and real-time inventory management decisions, enterprises can achieve more efficient and accurate supply chain management.

## CONCLUSION

In this article, we propose a TBN inventory management optimization framework based on causal inference Bayesian networks and transfer learning to address the inventory management optimization problem in supply chain management. This framework aims to achieve higher accuracy in inventory prediction and optimization within intelligent supply chain management systems. The framework predicts the remaining inventory using sales volume and sales quantity data and introduces a balanced adjustment of source and target domain parameters through balanced weight coefficients to optimize the model's transfer performance. Experimental results demonstrate that on the public dataset, the TBN framework achieves MSE and MAE values of 2.7 and 1.35, respectively, outperforming traditional machine learning methods such as SVM, MLP, and KNN. During the transfer process, the balanced coefficient-based transfer method surpasses improved Bayesian network methods like TAN and WATAN, delivering superior inventory prediction fitting analysis with simpler networks. Transfer analysis using both public and self-built datasets reveals that the TBN frameworks consistently yield better inventory forecasting results, effectively supporting the optimization and development of supply chain inventory management systems.

Future research will focus on further optimizing the model's dynamic adjustment capabilities to enhance the accuracy of inventory forecasting and optimization by integrating data from various dimensions, such as market dynamics and seasonal factors. Additionally, we aim to establish a standard supply chain inventory management research dataset to provide a reference for more researchers and business managers, thereby supporting the optimization and development of intelligent supply chain management systems.

# ACKNOWLEDGEMENTS

We thank the anonymous reviewers whose comments and suggestions helped to improve the manuscript.

## Funding

The authors received no funding for this work.

## Competing Interests

The authors declare that they have no competing interests.

## Author Contributions

- Zhu Xi conceived and designed the experiments, performed the experiments, performed the computation work, authored or reviewed drafts of the article, and approved the final draft.
- Wei Guan conceived and designed the experiments, performed the experiments, analyzed the data, prepared figures and/or tables, and approved the final draft.
- Ahmet Savasan performed the experiments, authored or reviewed drafts of the article, and approved the final draft.

## Data Availability

The Retail Sales Forecasting is available at Kaggle: https://www.kaggle.com/datasets/tevecsystems/retail-sales-forecasting.

The code is available in the Supplemental File.

## Supplemental Information

Supplemental information for this article can be found online at http://dx.doi.org/10.7717/peerj-cs.3262#supplemental-information.

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
