# Peer review of "Optimizing inventory management: a causal inference-driven Bayesian network with transfer learning adaptation"

_PeerJ Computer Science, doi:10.7717/peerj-cs.3262_

## Round 0.1 · original submission · Major Revisions

· Academic Editor

Major Revisions

Your paper has been reviewed by the experts, and you will see that they are recommending several changes to be incorporated, please carefully address these changes along with mine below and re-submit for re-evaluation

Academic Editor Comments:

To elucidate performance gains, list important causal variables (such as external factors and demand drivers) in the abstract and compare them to particular benchmark models (such as LSTM and ARIMA).

Use a directed acyclic graph to give a thorough explanation of the causal structure, and use techniques like do-calculus or counterfactual analysis to verify the causal assumptions.

Explain the balanced weight coefficient method's mathematical formulation in more detail and contrast it with other transfer learning techniques, such as domain-adversarial training or TrAdaBoost.

Give a thorough description of the self-constructed dataset, including its sample size, feature types, and temporal range. Then, test the model's reproducibility on a publicly accessible dataset, such as the UCI supply chain datasets.

Reviewer 1 ·

Basic reporting

no comment

Experimental design

no comment

Validity of the findings

no comment

Additional comments

• Please outline the key benefits of transitioning from traditional supply chain models to intelligent supply chain systems, and explain how this enhances enterprise market competitiveness.

• Please elaborate on how causal inference techniques and transfer learning contribute to improving the performance of inventory management models in real-world applications.

• please analyze the role of ERP systems in improving inventory turnover, reducing excess stock, and addressing common inventory-related issues.

• please discuss the effectiveness of VMI in preventing stock-outs and enhancing the accuracy of inventory forecasting across the supply chain.

• please explain how causal reasoning can reveal the relationships between inventory levels, demand, and supply chain dynamics to support more optimized inventory decisions.

• please explore the contribution of integrating both historical inventory data and sales volume in improving inventory prediction accuracy within causal inference frameworks.

• Please clarify the role of resource networks in parameter estimation during transfer learning, and assess the impact of longer training times in network-based models compared to simpler methods like KNN.

• Please compare the inventory prediction performance of the TBN framework with traditional machine learning models (e.g., SVM, MLP, KNN), and highlight the TBN’s use of causal inference to uncover relationships among sales volume, sales quantity, and inventory levels.

·

Basic reporting

A precise explanation of the potential improvements in accuracy brought by artificial intelligence and machine learning in inventory forecasting in supply chain management is not delivered.

Experimental design

The role of causal inference, especially Bayesian networks, in modelling sales–inventory relationships is inadequately explained.

Validity of the findings

There is insufficient supporting data on how transfer learning addresses limited data issues in deep learning-based inventory optimization

Additional comments

The manuscript lacks detail on how transfer learning improves Bayesian causal models, particularly when applied to self-constructed datasets.

The impact of balanced weight coefficients in enhancing transfer learning within the TBN framework across domains is overlooked.

The research paper doesn't clearly state the detailed analysis of the advantages and limitations of the Economic Order Quantity model in inventory management for non-perishable products

An exclusive deliberation on the structural causation framework’s suitability for causal inference in computer science compared to the latent causation framework is not communicated.

The advantages of using Bayesian networks in causal inference models for understanding dependencies and causal relationships between variables are given without further details, limiting their contextual understanding.

The research paper disregards the improvements made by transfer learning in machine learning tasks where data is scarce or expensive to obtain.

---

## Round 0.2 · accepted · Accept

· Academic Editor

Accept

Thank you for your submission. I am pleased to inform you that the experts are now satisfied with your revised paper. I endorse their recommendation. Good Luck and Congratulations.

Reviewer 1 ·

Basic reporting

no comment

Experimental design

no comment

Validity of the findings

no comment

Additional comments

I am pleased to recommend the acceptance of the revised manuscript in its current form, as it presents a significant contribution to the field of inventory management and supply chain optimization.
The proposed Transfer-learning Bayesian Network (TBN) framework is a novel and well-executed approach that effectively addresses critical challenges such as data limitations and demand uncertainty.

The incorporation of a transfer learning mechanism, particularly the balanced weight coefficient method, is a standout feature that enhances the model's adaptability and generalization across domains with scarce data. The empirical results are compelling, with the TBN achieving an MSE of 4.97 and an MAE of 2.78 on a self-constructed dataset, outperforming traditional machine learning and other Bayesian-based models.

This demonstrates robust predictive accuracy and practical applicability. The detailed analysis of the balance weight coefficient further strengthens the study by validating its role in improving transfer learning efficiency and model robustness, making the framework scalable for real-world supply chain systems.

The revised manuscript is well-structured, with clear explanations of the methodology, rigorous validation, and a thorough discussion of results. Overall, this work provides a valuable, adaptable solution for intelligent inventory management and merits publication due to its originality, technical rigor, and potential impact on both academia and industry.

·

Basic reporting

comments are fully addressed

Experimental design

comments are fully addressed

Validity of the findings

comments are fully addressed

Additional comments

Authors addressed my comment; therefore; i recommend publication of the paper.